# Precise engineering of quantum dot array coupling through their barrier widths

Ignacio Piquero-Zulaica[1], Jorge Lobo-Checa [2,3], Ali Sadeghi [4], Zakaria M. Abd El-Fattah [5], Chikahiko Mitsui[6], Toshihiro Okamoto[6,7], Rémy Pawlak [8], Tobias Meier [8], Andrés Arnau[1,9,10], J. Enrique Ortega[1,9,11], Jun Takeya [6], Stefan Goedecker[8], Ernst Meyer[8] & Shigeki Kawai [7,8,12]

Quantum dots are known to confine electrons within their structure. Whenever they periodically aggregate into arrays and cooperative interactions arise, novel quantum properties suitable for technological applications show up. Control over the potential barriers existing between neighboring quantum dots is therefore essential to alter their mutual crosstalk. Here we show that precise engineering of the barrier width can be experimentally achieved on surfaces by a single atom substitution in a haloaromatic compound, which in turn tunes the confinement properties through the degree of quantum dot intercoupling. We achieved this by generating self-assembled molecular nanoporous networks that confine the two-dimensional electron gas present at the surface. Indeed, these extended arrays form up on bulk surface and thin silver films alike, maintaining their overall interdot coupling. These findings pave the way to reach full control over two-dimensional electron gases by means of self-assembled molecular networks.

[1] Centro de Física de Materiales CSIC/UPV-EHU-Materials Physics Center, Manuel Lardizabal 5, E-20018 San Sebastián, Spain. [2] Instituto de Ciencia de Materiales de Aragón (ICMA), CSIC-Universidad de Zaragoza, E-50009 Zaragoza, Spain. [3] Departamento de Física de la Materia Condensada, Universidad de Zaragoza, E-50009 Zaragoza, Spain. [4] Department of Physics, Shahid Beheshti University, GC, Evin, 19839 Tehran, Iran. [5] Physics Department, Faculty of Science, Al-Azhar University, Nasr City E-11884 Cairo, Egypt. [6] Department of Advanced Materials Science, Graduate School of Frontier Sciences, The University of Tokyo, 5-1-5 Kashiwanoha, Kashiwa, Chiba 277-8561, Japan. [7] PRESTO, Japan Science and Technology Agency, 4-1-8, Honcho, Kawaguchi, Saitama 332-0012, Japan. [8] Department of Physics, University of Basel, Klingelbergstrasse 82, CH-4056 Basel, Switzerland. [9] Donostia International Physics Center (DIPC), Paseo Manuel Lardizabal 4, E-20018 Donostia-San Sebastián, Spain. [10] Dpto. de Física de Materiales, Universidad del País Vasco, E-20018 San Sebastián, Spain. [11] Dpto. Física Aplicada l, Universidad del País Vasco, E-20018 San Sebastián, Spain. [12] International Center for Materials Nanoarchitectonics, National Institute for Materials Science, 1-1, Namiki, Tsukuba, Ibaraki 305-0044, Japan. Correspondence and requests for materials should be addressed to J.L.-C. (email: jorge.lobo@csic.es) or to T.O. (email: tokamoto@k.u-tokyo.ac.jp) or to S.K. (email: Kawai.Shigeki@nims.go.jp)

Quantum dots (QDs) are analogous to artificial atoms as they confine electrons with discrete energy levels[1, 2]. They aggregate to form QD solids whose final properties are based upon their cooperative interaction, suitable for many technological applications[3–5]. Ideal QD solids demand truly monodisperse building blocks to prevent undesirable anomalies[3, 4, 6], but real ones exhibit significant structural variations. Digital structural fidelity is achieved on surfaces through atom-by-atom[7] and molecular manipulation[2] or by self-assembled molecular nanoporous networks[8–10]. Control of the potential barriers between neighboring QDs is essential to alter the crosstalk (interaction) between the existing units and engineer two-dimensional electron gases (2DEG)[8–15].

These self-assembled two-dimensional (2D) nanoporous networks are periodic extensions of quantum corrals[7, 16] that induce confinement of scattered surface 2DEG electrons inside its nanocavities[8–15]. Such regular nanostructures can be conceived as QD arrays[8, 17], where the surface adsorbed molecules (and adatoms) change the local surface potential landscape[11, 13, 18–20]. The confining barriers are characterized by a certain width and amplitude that affect the neighboring QDs' coupling degree[8, 17]. Electronic engineering of 2DEGs can be achieved by tuning the molecular building blocks, thereby altering the QD dimensions, the barrier amplitude and/or the barrier width[10, 20]. Moreover, changing the molecular compounds may also modify their interaction with the substrate[12], consequently altering the QD electron confinement strength. Whenever long-range regular structures are achieved, the surface 2DEG is modulated thereby generating new electronic bands whose dispersion relates to its interpore coupling strength and array dimensions[8, 11–13]. This long-range periodicity is desired for implementation into devices,

if bottom-up fabrication methods are used. However, the fine control in terms of the interpore distance (barrier width), while maintaining the potential barrier and pore dimensions is still elusive. In an ideal case where only the barrier width is disrupted, two paths can be envisioned to widen the interpore walls: the first, changing the length/width ratio of the molecule (preserving the overall interactions) and, the second, laterally stacking different number of constituents (altering the intermolecular and/or surface interactions).

In this work, we target this second path by means of two haloaromatic compounds that differ just in a single atom in their structure. Two different hexagonal molecular networks on a Ag (111) substrate are self-assembled that show single-molecular and double-molecular separation between their identical pores. The communication between neighboring QDs is investigated following the 2DEG modification through a combination of scanning tunneling microscopy/spectroscopy (STM/STS), angle resolved photoemission spectroscopy (ARPES) and extended model calculations.

## Results

**Atomic structure of single-wall and double-wall QD arrays.** Our concept to control the interpore barrier width while maintaining the pore size (Fig. 1a, b) is based upon the halogen bond versatility to generate artificial nanostructures. We employed two molecules (Fig. 1c, d): 3,9-dibromodinaphtho[2,3-b:2′,3′-d]thiophene (Br-DNT)[21] and [3,9-dibromodinaphtho[2,3-b:2′,3′-d] furan (Br-DNF). Intermolecular electrostatic attraction between the positive cap (σ hole) and the electron-rich regions (negative belt of bromine atoms[22] (Fig. 1c) or oxygen atom of the furan

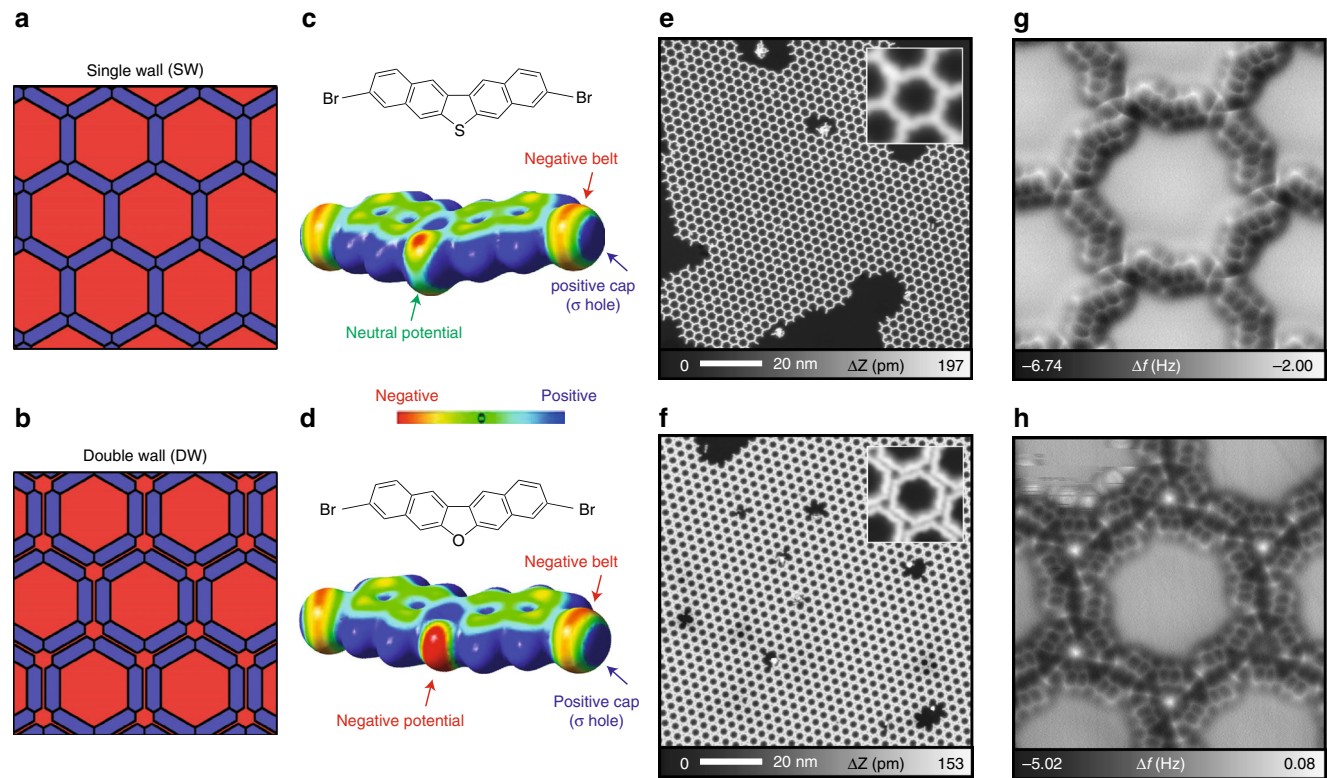

**Fig. 1** QD arrays generated by single-wall (SW) and double-wall (DW) nanoporous networks that confine the surface 2DEG. **a**, **b** Schematic representations of the concept behind SW and DW networks. **c**, **d** Chemical structures and electrostatic potential maps of Br-DNT and Br-DNF. **e**, **f** Large-scale STM topographies for the SW network with Br-DNT and the DW network with Br-DNF. Insets show close-views of each network. **g**, **h** High resolution atomic force microscopy (AFM) images of the SW network and DW network. Measurement parameters: tunneling current $I = 5$ pA, bias voltage $V = 200$ mV (**e**, **f**); $V = 0$ mV, oscillation amplitude $A = 60$ pm (**g**, **h**).

core (Fig. 1d)) are responsible for the condensation into two distinct molecular networks. Our STM images (Fig. 1e, f) show extended (over 100 nm) formation on Ag(111) with small amount of defects.

Detailed structures are derived using atomic force microscopy (AFM) with a CO functionalized tip[23]. The pores of the Br-DNT network are separated by a single molecule (Fig. 1g), whereas two are required for the Br-DNF network (Fig. 1h). Note that bright spots at the nodal sites of the latter are CO molecules adsorbed for tip functionalization (Supplementary Fig. 1). The condensation of Br-DNT happens solely through trigonal halogen bonding[24, 25], but the furan group presence in Br-DNF introduces higher interaction complexity. The O⋯Br-C bonding (oxygen is faintly observed in Supplementary Fig. 2) is apparently stronger (based on the electric potentials of Fig. 1c, d) than the Br⋯Br-C homo-halogen bond, leading to a shorter bond. Both hexagonal arrays are commensurate with the bare Ag(111) surface, according to density functional theory (DFT) calculations (Supplementary Fig. 3). Indeed, the Br-DNF network interpore distance is larger (by 14%) than that of Br-DNT, as a consequence of the molecular pairing. Note, however, that the enclosed pore areas remain identical to both assemblies (Supplementary Figs. 3 and 4). Essentially, we have structurally confirmed that these arrays can be conceived as extended model systems to investigate the 2DEG confinement and interpore coupling, as they feature identical QDs separated by different wall widths. For clarity, we will hereafter refer to the Br-DNT and Br-DNF networks as single-wall (SW) and double-wall (DW) networks, respectively.

**QD local electronic structure acquired by STS.** Figure 2a shows conductance ($dI/dV$) spectra acquired at the center of the pores of SW and DW networks and referenced to the clean Ag(111) substrate. Clear energy shifts of the pristine surface state onset (−65 meV) are induced by the QD confinement, peaking at 72 meV for the SW network and 45 meV for the DW network. These values are unexpectedly inverted since a stronger confinement (higher peak energy) is anticipated for the wider barrier (DW). It could be argued that it originates from a difference in the molecule-substrate interaction affecting the potential amplitude[20]. However, a larger interaction is expected for Br-DNF due to the extra oxygen electronegativity and the measured full width at half maximum of the confined state for the DW network (32 meV) is narrower than that for the SW network (45 meV). Moreover, the conductance maps at the peak energies show high conductance regions mainly located within the pores (Fig. 2b, c). Note that scarcely some pores present brighter contrast (indicated by arrows), which we assign to defects (Supplementary Fig. 5), and that the AFM images show both molecules lying flat on the surface (Fig. 1g, h and Supplementary Fig. 2).

**QD array electronic structure acquired by ARPES.** To understand the QD interactions and networks' confining properties, we performed ARPES measurements. This technique directly provides the electronic band structure of the systems, but endures the intrinsic restriction of being sensitive only to the occupied electronic structure[26–28]. Therefore, the confined states from the SW and DW networks on Ag(111) become undetectable. In order to push both confined STS peaks into the occupied region, we grew them onto a 3 monolayer (ML) Ag thin film on Au(111), which shifts the Ag(111) surface state by −100 meV while preserving its 2DEG character (Supplementary Figs. 6 and 7)[29]. The ARPES raw data (Fig. 3a–c) and corresponding second derivative (Fig. 3d–f), exhibit distinct network bands, characteristic of coupled QD arrays[8, 17]. These shift their minima to higher energy and deviate rapidly from the initial parabolic dispersions (weak side replicas away from $k_∥ ≃ ±0.1\ Å^{-1}$). The network band periodicities relate to their interpore distances and match our DFT calculations and STM data (cf. Table 1). Contrary to STS, we find that the band minimum shift (taking as reference the onset of the Ag layer) is larger for the DW than the SW network and exhibits a narrower bandwidth. This confirms a lower coupling between adjacent QDs for the DW case.

**EBEM modelisation for STS/ARPES comparison.** To validate a direct comparison between STS and ARPES results, we performed model calculations with the Electron Boundary Elements Method (EBEM)[11, 30]. We use simplified structures (straight beads) for the molecules (inset Fig. 4), which agree well with our DFT calculated iso-potential surfaces (Supplementary Fig. 3). In EBEM the Schrödinger equation is solved for independent electrons (2DEG) of effective mass $m^*$ within periodic domains containing two different potential areas: zero for Ag sites (pores and substrate) and constant non-zero potential ($V_{eff}$) for the molecular positions (walls) that scatter the electrons. The $m^*$ and $V_{eff}$ parameters are determined by an iterative fitting using both STS and ARPES data (for details see EBEM simulations section in Methods and Supplementary Methods sections). The agreement of these calculations turns out to be excellent, confirming that both substrates (bulk and thin film) are equivalent for our study. The only requirement is a 100 meV shift to account for the different 2DEG onset. We obtain a common $V_{eff}$ of 140 meV at the molecular sites, but different effective masses: $m^*_{Ag} = 0.38\ m_0$, $m^*_{SW} = 0.49\ m_0$ and $m^*_{DW} = 0.54\ m_0$. Such effective mass increase will be discussed later, but suggests a change in the electron wavefunction overlap

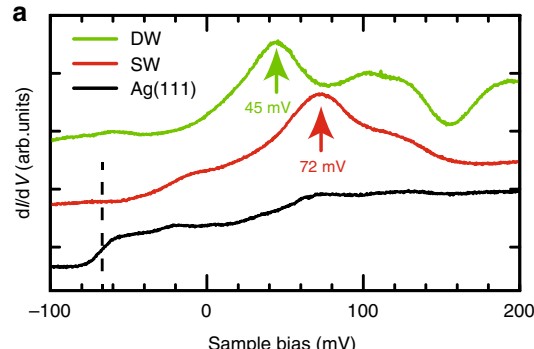

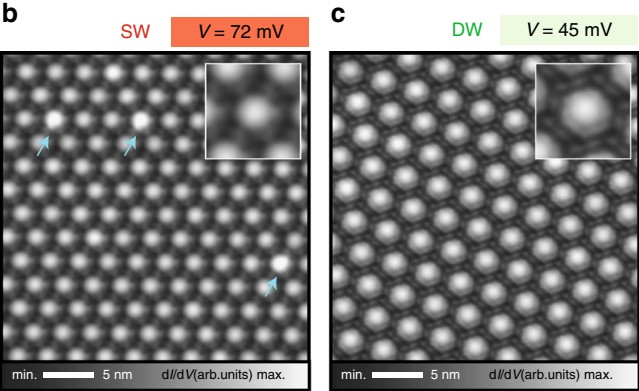

**Fig. 2** Local electronic structure of the SW and DW networks. **a** Conductance ($dI/dV$) spectra on the pristine Ag(111) (*black*), on the pore centers of the SW network (*red*) and DW network (*green*). **b**, **c** $dI/dV$ maps on the SW and DW networks acquired on the peak maxima in **a**, enlarged in the corresponding insets. Measurement parameters: $I = 10\ pA$, $V = 72\ mV$, modulation voltage $V_{ac} = 10\ mV$, oscillation frequency $f_{ac} = 513\ Hz$ (**b**); $I = 10\ pA$, $V = 45\ mV$, $V_{ac} = 10\ mV$, $f_{ac} = 515\ Hz$ (**c**).

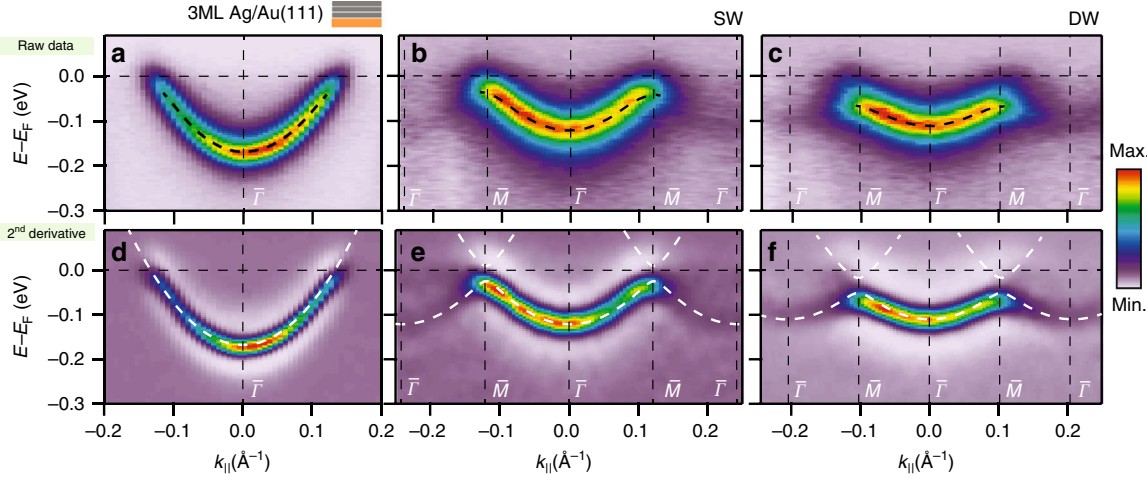

**Fig. 3** 2DEG modification induced by the SW and DW network potential barriers. **a–c** ARPES map along $\overline{\Gamma M}$, obtained on the 3ML Ag/Au(111) as well as on the SW and DW networks. The energy distribution curves close to $\overline{\Gamma}$ were fitted, using a Lorentzian component and a linear background convoluted with a Fermi function (black dashed lines). **d–f** Second derivative maps of the above raw data for an improved visualization of the second surface Brillouin Zone. The white dashed lines correspond to the EBEM calculated electronic bands stemming from altering the 2DEG with the molecular surface potentials.

**Table 1 Extracted ARPES experimental parameters from the bands presented in Fig. 3**

|  | 3 ML Ag film | SW/Ag film | DW/Ag film |
|---|---|---|---|
| Band bottom | −160 meV | −120 meV | −110 meV |
| Band width | — | 92 meV | 51 meV |
| $\overline{M}$ point | — | 0.120 Å$^{-1}$ | 0.104 Å$^{-1}$ |
| Interpore distance | — | 3.02 nm | 3.49 nm |
|  |  | (3.03 nm) | (3.45 nm) |
| $m^*/m_0$ | 0.38 | 0.47 | 0.59 |

The interpore distance match, within the experimental error, that obtained by DFT calculations (values in parentheses)

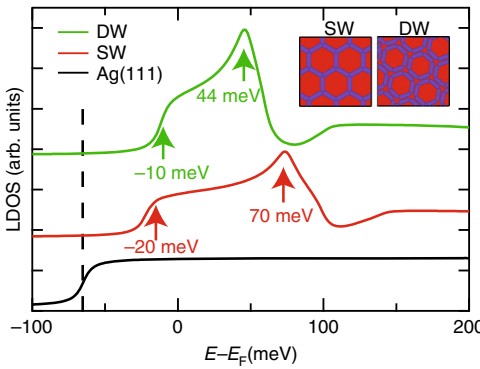

**Fig. 4** EBEM simulations of the local electronic structure. Calculated d$I$/d$V$ spectra, obtained after fitting the experimental data with EBEM. The ARPES fit was shifted by 100 meV for direct comparison with the d$I$/d$V$ spectra in Fig. 2a. Insets show both molecular geometries used in EBEM calculations that closely match the STM topographies (Fig. 1e, f) and DFT calculated electric field profiles (Supplementary Fig. 3).

with the crystal substrate concomitant to an enhancement of the pore confinement leading to a reduction of the QD coupling when going from SW to DW (Supplementary Fig. 8). The reproduced band structures (white dashed lines in Fig. 3d–f) match exceptionally well our ARPES data (Supplementary Fig. 9), validating the expected pore coupling difference and periodic

long-range order of these arrays. Moreover, the calculated local electronic structure (Fig. 4) also agrees with the STS data of Fig. 2a not only in the peak values but also in shape. Therefore, the EBEM simulations bring conclusive consistency when comparing the experimental STS on Ag(111) and ARPES on 3ML Ag/Au(111).

## Discussion

The inverted order of the STS energy peaks can now be explained. Coupled QDs give rise to bonding and anti-bonding continuum states when set in arrays[19]. The fundamental energy is established by the bonding state and the overall bandwidth (proportional to the QD interaction) is limited by the anti-bonding ones. The reduced bandwidth of the DW network compared to the SW (by ~45%, cf. Table 1) confirms the lower interpore coupling imposed by the wider barriers. However, this does not explain the larger STS peak shifts at the pore center with respect to the ARPES fundamental energies. The underlying reason is that the STS technique reveals an enhanced sensitivity to probe the anti-bonding state[17, 19]. According to Seufert et al.[19] the wavefunction shape for the bonding state is more spread out than the anti-bonding one. Thus, the latter peaks more abruptly at the pore center, yielding a higher conductance than the former (for a particular tip height). Consequently, the peak lineshapes are generally asymmetric with maxima displaced towards the top of the band (Figs. 2a and 4), which in ARPES matches the $\overline{M}$ point energy (after shifting 100 meV). Nevertheless, the STS is still sensitive enough to the fundamental energy (bonding state) as they sometimes appear as onsets in the spectra. In particular, these can be deduced from Fig. 4 (at −20 meV for the SW and −10 meV for the DW) and match the ARPES energy minima ($\overline{\Gamma}$ point). Therefore, the bandwidth and corresponding QD interaction could be estimated from the STS peak width whenever sharp cutoffs show up in the spectra.

The value of $V_{\text{eff}} = 140$ meV from our model calculation yields an overall barrier of $V_{\text{eff}} \cdot d = 0.7$ eVÅ per molecule, which is small compared to other networks[9, 11]. Such weak potential barrier can originate from the weak interaction between the haloaromatic compound and the substrate or by the absence of metallic coordination in our arrays. Moreover, the use of solely the substrate's 2DEG $m^*$ does not provide a good agreement for the networks (Supplementary Fig. 8) and we must recurrently use an increased value close to the experimental one. This suggests that,

besides the lateral scattering at the molecule network, there is a subtle change in the electron wavefunction overlap with the crystal substrate. Note, however, that we expect this vertical overlap to be practically identical for both networks, given that $V_{eff}$ is the same. Therefore, the additional increase of $m^*$ when going from SW to DW barriers, with associated flattening of the bands, suggests a correlation with QD intercoupling. In essence, the $m^*$ increase and band flattening could be considered like fingerprints for increased electron localization and reduced interdot coupling.

In summary, our work shows that precise engineering of QD array coupling is possible by modifying just the barrier width (without affecting QDs' size). These organic nanoporous networks are generated on bulk and thin Ag films alike by substitution of a single atom in the precursor molecule, reminiscent of a butterfly effect. The extended and periodic nature of these arrays provides access to their distinct band structures, which are directly compared with their local density of states and merged through calculations. Such complementary experimental and theoretical synergy provides complete fundamental insight into the nature of QD intercoupling processes. Even though substrate contributions cannot be discarded, our findings clearly suggest that the reduction of the QD coupling (from SW to DW) is associated with a flattening of the band dispersion and increase of the effective mass. This work not only complements the toolbox for tuning surface electronic properties, which started in the 90's with the quantum corrals[7], but it is also prone to help in deriving clear conceptual ideas on QD coupling, which is an essential parameter for next-generation computing or device technologies.

## Methods

**STM/AFM measurements.** All experiments were performed with Omicron STM/AFM with a qPlus configuration[31], operating at 4.8 K in UHV. A clean Ag(111) surface was in-situ prepared by repeated cycles of standard sputtering and annealing. The W tip of a tuning fork sensor was ex-situ sharpened by focused ion beam milling technique and was then in-situ covered with Ag atoms by contacting to the sample surface.

3,9-dibromodinaphtho[2,3-b:2′,3′-d]thiophene (Br-DNT)[21] and 3,9-dibromodinaphtho[2,3-b:2′,3′-d]furan (Br-DNF) (synthesis description in the Supplementary Methods) were deposited on Ag(111) surfaces at 150 K from a crucible of Knudsen cell. The resonance frequency of the self-oscillating qPlus sensor was detected by a digital lock-in amplifier (Nanonis: OC4 and Zurich Instruments: HF2LI and PLL). In STM mode, the tip was biased while the sample was electronically grounded. The topographic images were taken in a constant current mode. In AFM mode, the tip apex was terminated by a CO molecule[23] and all images were taken at a constant height mode.

**ARPES experiments.** Our home laboratory angle resolved photoemission (ARPES) setup consists of a display type hemispherical analyzer (Phoibos150) with an energy/angle resolution of 40 meV/0.1° and a monochromatized source He I$_\alpha$ (h$\nu$ = 21.2 eV) source. The channel plate slit lies along the rotation axis of the manipulator. All the presented data were recorded approximately at 130 K.

A clean Au(111) surface was in-situ prepared by repeated cycles of standard sputtering and annealing. To form Ag films of controlled monolayer thickness on Au(111), a wedge-like mask was positioned in front of the substrate and was moved slowly during the Ag deposition at 150 K[32]. Afterwards, the sample was heated up to ~450 K to improve the surface quality. Br-DNT and Br-DNF were deposited on the substrate at ~130 K. After each preparation step, we controlled the sample quality by measuring its electronic structure.

**EBEM simulations.** The combined Plane Wave Expansion (PWE) and Electron Boundary Element Method (EBEM) have been developed by García de Abajo and represents a scalar variant of the electromagnetic PWE/BEM extensively used for solving Maxwell's equations and optical response for arbitrary shapes. It is based on Green's functions for finite geometries and electron plane wave expansion for periodic systems. For the band structure calculations, the particle-in-a-box model is extended to infinite 2D systems by defining an elementary cell and using periodic boundary conditions. Within the PWE code, solutions of the Schrödinger equation are represented as a linear combination of plane waves and a satisfactory convergence was achieved with a basis set consisting of ~100 waves.

**Ab initio calculations.** The calculations were carried out within the local density approximation of density functional theory (DFT) as implemented in the BigDFT code[33]. A wavelet basis set was used to expand the wavefuntion of the valence electrons while the core electrons were removed using norm-conserving HGH pseudopotentials[34]. Calculating the electrostatic potential (and the electric field) for a surface system is uniquely precise by the Poisson solver of this DFT code that allows to apply periodic boundary conditions along two in-plane directions while keeping free boundary conditions out of the plane direction[35].

**Data availability.** The data that support the findings of this study are available from the corresponding authors on request.

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

## Acknowledgements

We are grateful to J. García de Abajo for providing the EBEM code. I.P.-Z. and J.L.-C. thank G. Sauthier for ARPES technical assistance. This work was supported in part by the Spanish Ministry of Economy (grants MAT2013-46593-C6-4-P, MAT2016-78293-C6-6-R and FIS2013-48286-C2-1-P), by the Spanish Research Council (CSIC- 201560I022), by the Basque Government (grants IT621-13 and IT-756-13), by the Japan Science and Technology Agency (JST), 'Precursory Research for Embryonic Science and Technology' (PRESTO) for a project of 'Molecular technology and creation of new function', by JSPS KAKENHI Grant Number 15K21765, by the Swiss National Science Foundation, by the Swiss Nanoscience Institute, and by COST Action (European Cooperation in Science and Technology) MP1303. Swiss National Supercomputing Center in Lugano (Project s499 and s621) is acknowledged.

## Author contributions

S.K. conducted the STM/STS and AFM measurements and its data analysis; I.P.-Z. and J.L.-C. conducted the ARPES experiments and corresponding data analysis; Z.M.A.E.-F. conducted the EBEM simulations; A.S. performed the DFT calculations; C.M., T.O. and J.T. synthesized and purified the precursor molecules; J.L.-C., S.K., A.S., I.P.-Z. and T.O. contributed to writing the manuscript. All authors contributed to the revision and final discussion of the manuscript; J.L.-C. and S.K. conceived this project.

## Additional information

**Competing interests:** The authors declare no competing financial interests.

