## [transparent peer review File · Nature Communications]

Reviewer #1 (Remarks to the Author):

Authors present an elegant study on the electron confinement properties through different molecular arrays with nanoporous character on silver bulk and thin films samples with a (111) orientation. The employed methodology, including tunneling and angle-resolved photoemission spectroscopy is pertinent, and complemented by model simulations. A key finding is that slightly different building units of network structures correlate with different confinement properties and that the respective quantum dot arrays thus can be (in principle) precisely controlled - provided a detailed understanding of the underlying physics can be elaborated.

The ms is well written with the experimental and theoretical work performed in a very competent way. Data of high quality. The study also addresses questions of high current interest, though authors do not provide adequate information concerning some speculative statements related to the application potential of the realized systems, seemingly providing 'the core of non-standard computers' or being useful for 'implementation in future devices' etc.. Interfacial electron confinement has been extensively studied, with a main push achieved by SPM techniques a long time ago. The present study continues a nice line of research, recognizable in the series of cited papers, but it is not quite clear in which way the specific findings, which are certainly of high scientific quality, could propel computing or device technologies. This point requires better justification and explanation or otherwise the statements are to be toned down.

- p. 3: 'paths can be envisioned'... it is also clear from the beginning that surface interactions and constitution of molecular building blocks are decisive.

- p. 6: determination of M^* and V_{eff} by 'iterative fitting' does not say much - how could an expert in the field redo the procedure or verify it?

- p. 6: reasons / discussion for different m^* should be provided

- how can the findings be generalized to realize QD arrays in a predictive way and with predictive properties?

What is the relevance of the QD coupling effects, nicely analyzed presently in this context? Can one derive clear conceptual ideas to reach next-generation computing or device technologies?

- conclusion to be strengthened / generalized. Also explain how scientific understanding could influence technology.

Reviewer #2 (Remarks to the Author):

Piquero-Zulaica et al reported a combined study of the surface state quantum dot (QD) formed in self-assembled network of two molecules Br-DNT and Br-DNF using ARPES, STS and EBEM calculations. They found that Br-DNT formed single wall (SW) network while Br-DNF formed double wall (DW) networks on Ag(111). The two networks thus possess pores with identical size but different pore-to-pore distance, providing a good system to understand the effects of QD coupling. The STS data show counter-intuitive results: the half maximum of the confined state of SW network is broader than DW network, while the peak of the confined state of SW network shifts more strongly than DW. To confirm this trend, the authors further grew the same networks on a Ag/Au(111) surface and used ARPES to resolve the band structures. The results are consistent with STS, showing that the band bottom of the SW network shifts more than DW. The counter-intuitive shifting of the QD states is explained by the enhanced sensitivity of the antibonding state.

Overall, the results presented in this manuscript are completely new and would have impact on the future studies on this direction. The paper is well written, the data are in very high quality and the data analysis and theoretical calculation support the conclusion except the comment below. I

recommend the publication of this paper in Nature Communication after the comments below being satisfactorily addressed.

Comments:

1. ARPES data were acquired on a Ag/Au(111) substrate while STM/STS data were acquired on Ag(111). To be conclusive, the authors shall provide STM/STS data of the both networks grown on Ag/Au(111) surface, either in the main manuscript or in the supporting information.
2. The explanation of the anti-bonding states is very brief: "The underlying reason is that the STS technique reveals an enhanced sensitivity to probe the antibonding state" on page 7. The authors shall elaborate this point in more details since this is the major discovery of this work.

Reviewer 1

We thank Reviewer 1 because the points he/she has raised are certainly relevant leading to a general improvement of our manuscript, although some comments lead to some speculation from our side:

1.- The study also addresses questions of high current interest, though authors do not provide adequate information concerning some speculative statements related to the application potential of the realized systems, seemingly providing 'the core of non-standard computers' or being useful for 'implementation in future devices' etc.. Interfacial electron confinement has been extensively studied, with a main push achieved by SPM techniques a long time ago. The present study continues a nice line of research, recognizable in the series of cited papers, but it is not quite clear in which way the specific findings, which are certainly of high scientific quality, could propel computing or device technologies. This point requires better justification and explanation or otherwise the statements are to be toned down.

After the Reviewer's comments, we have realized that these two statements are vague and could be misleading. Thus, we have decided to follow the advice and toned down our statements. Having said that, we believe that our work can serve as conceptual playground if ever quantum dot arrays become implemented into technological devices (read replies to his/her following comments).

We remove the sentence related to computers, as it does not affect the message of the manuscript. The other sentence ("This long-range periodicity is key for prospective implementation into future devices") is modified since we believe it is correct in the context of bottom up fabrication: *"This long-range periodicity is desired for implementation into devices, if bottom-up fabrication methods are used"*.

2.- p. 3: 'paths can be envisioned'... it is also clear from the beginning that surface interactions and constitution of molecular building blocks are decisive.

We agree with the Reviewer that the molecular interactions and constitution of molecular building blocks are truly relevant so our previous sentence was not precise. Thus, we have modified it to clarify that we are considering an "ideal case" (only adjusting the barrier width *without affecting the barrier height or pore size*) and we explicitly indicate that the molecule and surface interactions must be modified in one case but not in the other:

"In an ideal case where only the barrier width is disrupted, two paths can be envisioned to widen the interpore walls: the first, changing the length/width ratio of the molecule (preserving the overall interactions) and, the second, laterally stacking different number of constituents (altering the intermolecular and/or surface interactions)."

3.- p. 6: determination of M^ and V_{eff} by 'iterative fitting' does not say much - how could an expert in the field redo the procedure or verify it?*

We agree with the Reviewer that no details in this respect were provided, as previous references contain this information. In essence, for the modellization using EBEM we start parametrizing the network geometry. Once this is set, the fitting of the experimental data depends on two parameters: the effective molecular barrier potential (V_{eff}) and the effective mass (m^*). With V_{eff} , the scattering of the surface state with the molecular barriers is defined and both the band bottom (Gamma point in ARPES and onset of bonding state in STS) and the opening of the gaps are quantitatively determined. The effective mass parameter is inversely proportional to the dispersion of the band so the curvature, i.e. the top of the band or antibonding state, is obtained. Please note that changes in m^* do not alter significantly the energetic position of the band bottom of the first state. Thus, by tweaking these two parameters we can play until we get a perfect agreement with the experimental data.

This discussion has been included in the Supplementary Information, in the "EBEM simulation" section. We refer the reader to it in the main text: *"The m^* and V_{eff} parameters are determined by an iterative fitting using both STS and ARPES data (for details see EBEM simulations section in the Supplementary Information)."*

4.- p. 6: reasons / discussion for different m^ should be provided*

The underlying reasons why m^* shows such differences are not straightforward. The effective mass of surface states reflect the way the surface electron interacts with the substrate. First, the surface electronic state adopts a similar effective mass value from the nearest bulk band at the zone center, so shifting it away from that band edge will modify its m^* as it approaches the other band gap edge (check for image states in Phys. Rev. B 35, 975 (1987)). In this particular case (upward shift of a Shockley state), it results as an increase of m^* . Moreover, shifting up the bottom energy spatially pushes the surface state closer to the vacuum (away from the substrate), thus becoming more free-electron like (m^* increases to unity (m_0)). This discussion was addressed in p.8 of original manuscript with the sentences: *"The use of solely the substrate's 2DEG m^* does not provide a good agreement for the networks and we must recurrently use an increased value close to the experimental one. This suggests that, besides the lateral scattering at the molecule network, there is a subtle change in the electron wavefunction overlap with the crystal substrate"*.

Despite this, we believe there must also be a correlation with the lateral QD coupling. In particular, we expect SW and DW to have practically identical wavefunction overlaps with the substrate, since V_{eff} is the same for both. However, the marked flattening of the bands and increased m^* from SW to DW can only be explained, in our opinion, by an increased electronic localization and reduced interdot coupling. The extreme case leading to total confinement (no QD overlap) would yield an infinite m^* , i.e. complete flattening of the bands.

We admit that we are a bit speculative in our arguments, since we cannot out rule the substrate's contribution, but following the Reviewer's advice (and later comments), we will introduce this in the manuscript. The following sentences are aggregated:

- (After the values are listed from the EBEM modelisation, p.7): *Such effective mass increase will be discussed later, but suggests a change in the electron wavefunction overlap with the crystal substrate concomitant to an enhancement of the pore confinement leading to a reduction of the QD coupling when going from SW to DW (see Supplementary Fig. 8).*

- (At the end of the discussion (1st paragraph of p. 9): *Note however that we expect this vertical overlap to be practically identical for both networks, given that V_{eff} is the same. Therefore, the additional increase of m^* when going from SW to DW barriers, with associated flattening of the bands, suggests a correlation with QD intercoupling. In essence, the m^* increase and band flattening could be considered as fingerprints for increased electron localization and reduced interdot coupling.*

5.- how can the findings be generalized to realize QD arrays in a predictive way and with predictive properties?

We feel that addressing this question is far too speculative since we have always experienced that organic nanoporous networks formation on surfaces is partly random. For instance, exchanging the noble metal substrate from Ag(111) to Au(111) dramatically hinders the formation of these networks. This evidences that there is a delicate balance in the self-assembly processes, where molecule-substrate interactions (diffusion energy, kinetic effects) and intermolecular interactions (self-healing bonding mechanism or one-way covalent bonding) must be collectively taken into account (see for instance L. Porte, PRB 84, 125421 (2011)). We can only foresee that, whenever porous networks are obtained, determining the scattering molecular barrier strength becomes essential for predicting the confinement properties of 2DEGs, as weakly scattering systems deviate from an infinite well particle in a box model. Thus, we refrain from generalizing or using our findings in a predictive way as we consider this highly risky.

6.- What is the relevance of the QD coupling effects, nicely analyzed presently in this context? Can one derive clear conceptual ideas to reach next-generation computing or device technologies?

Following the first point raised by the Reviewer, we decided to minimize the speculative arguments in our manuscript. However, here we did show how the molecular barriers do actively control the degree of QD coupling. We have (after this Reviewing process) highlighted that modification of the coupling is likely associated to direct changes of the effective mass or the corresponding band dispersion that cannot be solely explained in terms of lateral scattering at the molecular barriers. It has been postulated that the modulation (coherence) is of utmost importance for the Quantum bits stability in semiconducting materials (see Loss-DiVincenzo quantum computer) and also that “the qubits must interact very *strongly* with one another to make logic gates and transfer information” (Christopher Monroe, “Quantum Computing”, PMID: PMC33891). Therefore, we will add a closing sentence with the idea that our work can

help to derive clear conceptual ideas to reach next-generation computing or device technologies in the closing paragraph:

"This work... [...] is also prone to help in deriving clear conceptual ideas on QD coupling, which is an essential parameter for next-generation computing or device technologies"

7.- Conclusion to be strengthened / generalized. Also explain how scientific understanding could influence technology.

The Reviewer is correct that we did not include a proper conclusion paragraph. Following his/her advice, we add the following paragraph starting at the end of page 9:

"In summary, our work shows that precise engineering of QD array coupling is possible by modifying just the barrier width (without affecting QDs' size). These organic nanoporous networks are generated on bulk and thin Ag films alike by substitution of a single atom in the precursor molecule, reminiscent of a butterfly effect. The extended and periodic nature of these arrays provides access to their distinct band structures, which are directly compared with their local density of states and merged through calculations. Such complementary experimental and theoretical synergy provides complete fundamental insight into the nature of QD intercoupling processes. Even though substrate contributions cannot be discarded, our findings clearly suggest that the reduction of the QD coupling (from SW to DW) is associated with a flattening of the band dispersion and increase of the effective mass. This work not only completes the toolbox for tuning surface electronic properties, which started in the 90's with the quantum corrals, but it is also prone to help in deriving clear conceptual ideas on QD coupling, which is an essential parameter for next-generation computing or device technologies."

Reviewer 2

The results presented in this manuscript are completely new and would have impact on the future studies on this direction. The paper is well written, the data are in very high quality and the data analysis and theoretical calculation support the conclusion except the comment below. I recommend the publication of this paper in Nature Communication after the comments below being satisfactorily addressed.

We thank Reviewer 2 for his/her excellent consideration of our work and for his/her recommendation in favor of publication. In the following, we address the raised comments:

1.- ARPES data were acquired on a Ag/Au(111) substrate while STM/STS data were acquired on Ag(111). To be conclusive, the authors shall provide STM/STS data of the both networks grown on Ag/Au(111) surface, either in the main manuscript or in the supporting information.

We consider that, in order to be conclusive, no new STS measurements are needed on the networks grown on Ag/Au(111). When dealing with electronic states, ARPES is complementary to STS, but can at times be advantageous for the following reasons:

(i) There are no uncertainties introduced by external probing elements (tip in STM). In STS the measured electronic states emerge as a convolution with the density of states of the tip, which introduce many ambiguities in the spectra. These are absent in ARPES.

(ii) There are practically no perturbations to the overall electronic signal from localized defects. STM probes at the local scale, so any molecular and/or substrate modification will visibly change the conductance spectra. Conversely, ARPES is a space averaging technique and only the periodically repeated and identical (defectless) units will contribute to the signal.

When QD intercoupling is resolved by ARPES for these networks, it can (*and does here*) shed light upon existing controversies in the observed STS conductance, as highlighted by Reviewer 2 in the first paragraph from his/her review. The reason why ARPES has practically not been used to study the electronic response of QD arrays is that ARPES demands practically perfect sample quality conditions, which is not as critical when performing STM/STS. From our experience, to obtain unambiguous electronic band structures from porous networks (such as the one shown here), the systems must fulfill simultaneously all of these highly demanding requirements: (i) laterally extended regular networks, (ii) practically monodomain structures, (iii) very small amount of defects, and (iv) large presence of the network on the surface (practically filling it). Indeed, the preparation and choice of systems is so critical that to date published work showing evidence of experimental ARPES band structures have been only achieved in a molecular network and by part of these Authors (Refs. 8, 17 and 32). Note that failed ARPES attempts have been even reported (see for instance fig. 3 of Ref. 14).

Thus, in sight of the unambiguous match of STM and ARPES periodicity, and the exceptional agreement derived simultaneously from the EBEM calculations (note that the simulations use exactly the same geometries and fitting parameters) we must conclude that the systems are identical demonstrating the validity of our results. Showing any more STS data will not (and cannot) provide additional evidence or spectroscopic findings to what is already presented here.

However, inspired by the Reviewer's comment, we have added a sentence in the closing paragraph (page 9) declaring that the conclusiveness of our results resides precisely in the complementarity of the experimental techniques and the theoretical simulations: "*Such complementary experimental and theoretical synergy provides complete fundamental insight into the nature of QD intercoupling processes.*"

2.- The explanation of the anti-bonding states is very brief: "The underlying reason is that the STS technique reveals an enhanced sensitivity to probe the antibonding state" on page 7. The authors shall elaborate this point in more details since this is the major discovery of this work.

We agree with the Reviewer that the text dealing with the discussion of bonding-antibonding states that compare the ARPES and STS signals is rather brief. We initially considered that this quite important result was already well described and reported by Seufert et al. in Ref. 19 (text top of Fig. 3): *"The bonding state, showing up at lower energy, has larger probability density inside the central barrier ($V = V_0$), while the higher energy, antibonding state exhibits a node in the central barrier, accompanied by an increase in probability density in the outer walls ($V = 0$)"*. In essence, Seufert et al. state that the antibonding states have a higher probability or weight at the pore center because of a stronger localization of its wavefunction at that position, as compared to the bonding state. Thus, the STM tip will access (at the pore center) the anti-bonding state more effectively than the other, generating the asymmetry in the conductance.

Thus, as requested by the Reviewer, we have further elaborated this point in the manuscript by adding the following sentences (page 8): *"According to Seufert et al. the wavefunction shape for the bonding state is more spread out than the anti-bonding one. Indeed, the latter peaks more abruptly at the pore center, yielding a higher conductance than the former (for a particular tip height)"*.

Changes introduced into the manuscript and supplementary Information

All changes have been *highlighted* in the main text and S.I. based on the advice from both Reviewers and following the checklist provided by the journal.

These are the **changes due to the Reviewers comments**:

- Modification of two sentences in the second paragraph of the introduction following the advice from comments 1 and 2 of Reviewer 1 (end of page 3).
- Sentences added in pages 7 and 9 following comment 4 from Reviewer 1.
- Include discussion elaborating the reasons for the enhanced STS sensitivity to the anti-bonding state (page 8), as requested by Reviewer 2.
- Extend the final paragraph before the materials and methods to strengthen and generalise the conclusions (2nd paragraph of page 9), following the last comment from Reviewer 1. Note that a sentence is added in this paragraph following first comment from Reviewer 2.
- The paragraph of the "EBEM simulation" has been extended in the S.I. to include a discussion of the iterative fitting of V_{eff} and m^* , as requested by Reviewer 1 in his 3rd comment.

These are the **changes due to the Editorial checklist**:

- New abstract. We have re-written a shorter version of the abstract to comply with the Journal editorial line (150 words max. length and no references). Accordingly, the first paragraph of the initial manuscript has been split into two parts now included in the

abstract (from "Here we show...") and as the start of the introduction paragraph (not modified).

- Modification of the first sentence of the last paragraph of the introduction (beginning of page 4) to comply with the editorial checklist that demands a brief summary of the results and the conclusions.
- Provide structure to the text adding sections (Results and Discussion) and subheadings.
- Refer to Supplementary Information figures using the whole word (Supplementary).
- Include Methods section after the Discussion Section. This text has been shifted from the previous Supplementary Information.
- Include the Author contribution.
- The format of the Supplementary Information has been changed to a Word file.

We thank again both Reviewers and sincerely appreciate their help for improving the overall quality of our manuscript.

Best regards,

Jorge Lobo-Checa (on behalf of all co-Authors)

Reviewer #1 (Remarks to the Author):

[The referee believes the manuscript is now ready for publication with no further comments for the authors apart from noting that, in the summary sentence, 'work not only completes the toolbox for tuning surface electronic properties ... ' -> 'work not only complements the toolbox for tuning surface electronic properties...']

Reviewer #2 (Remarks to the Author):

The revised manuscript and the reply nicely address my comments. The manuscript is publishable.

Reviewer #1:

The referee believes the manuscript is now ready for publication with no further comments for the authors apart from noting that, in the summary sentence, 'work not only completes the toolbox for tuning surface electronic properties ... ' -> 'work not only complements the toolbox for tuning surface electronic properties...'

We thank Reviewer #1 for the positive assessment of our work. We have followed his/her advice and have changed the sentence in the last paragraph of the discussion so that it now reads: "This work not only **complements** the toolbox..."

Reviewer #2:

The revised manuscript and the reply nicely address my comments. The manuscript is publishable.

We are grateful to Reviewer #2 for his/her positive consideration of our manuscript and recommendation to publish it in Nature Communications.

Best regards,

Jorge Lobo-Checa (on behalf of all co-Authors)